# The Presence of Either Typical or Atypical Radiological Changes Predicts Poor COVID-19 Outcomes in HIV-Positive Patients from a Multinational Observational Study: Data from Euroguidelines in Central and Eastern Europe Network Group

**DOI:** 10.3390/v14050972

**Published:** 2022-05-05

**Authors:** Justyna D. Kowalska, Carlo Bieńkowski, Lukáš Fleischhans, Sergii Antoniak, Agata Skrzat-Klapaczyńska, Magdalena Suchacz, Nikolina Bogdanic, Deniz Gokengin, Cristiana Oprea, Igor Karpov, Kerstin Kase, Raimonda Matulionyte, Antonios Papadopoulos, Nino Rukhadze, Arjan Harxhi, David Jilich, Botond Lakatos, Dalibor Sedlacek, Gordana Dragovic, Marta Vasylyev, Antonia Verhaz, Nina Yancheva, Josip Begovac, Andrzej Horban

**Affiliations:** 1Department of Adults’ Infectious Diseases, Hospital for Infectious Diseases, Medical University of Warsaw, 01-201 Warsaw, Poland; jdkowalska@gmail.com (J.D.K.); agata.skrzatasw@gmail.com (A.S.-K.); ahorban@zakazny.pl (A.H.); 2Department of Infectious Diseases, 1st Faculty of Medicine, Faculty Hospital Bulovka Hospital, Charles University in Prague, 18081 Prague, Czech Republic; lukas.fleischhans@seznam.cz (L.F.); david.jilich@centrum.cz (D.J.); 3Viral Hepatitis and AIDS Department, Gromashevsky Institute of Epidemiology and Infectious Diseases, Amosova str. 5-a, 03038 Kyiv, Ukraine; antoniakserge@gmail.com; 4Department of Infectious and Tropical Diseases and Hepatology, Medical University of Warsaw, 01-201 Warsaw, Poland; magdalena.suchacz@wum.edu.pl; 5School of Medicine, University Hospital for Infectious Diseases, University of Zagreb, Miorogojska 8, 10000 Zagreb, Croatia; nikolinabogdanic@gmail.com (N.B.); josip.begovac@gmail.com (J.B.); 6Department of Infectious Diseases and Clinical Microbiology, Faculty of Medicine, Ege University, 35100 Izmir, Turkey; gkengin61@gmail.com; 7Victor Babes Clinical Hospital for Infectious and Tropical Diseases, Carol Davila University of Medicine and Pharmacy, 030303 Bucharest, Romania; crisoprea2512@yahoo.com; 8Department of Infectious Diseases, Belarusian State Medical University, Dzerginskogo 83, 220116 Minsk, Belarus; vip.kia1957@mail.ru; 9West Tallinn Central Hospital, Paldiski Road 62, 10149 Tallin, Estonia; doktorkase@gmail.com; 10Faculty of Medicine, Vilnius University Hospital Santaros Klinikos, Vilnius University, 08410 Vilnius, Lithuania; raimonda.matulionyte@santa.lt; 11Medical School, University General Hospital Attikon, National and Kapodistrian University of Athens, 12462 Athens, Greece; antpapa1@otenet.gr; 12Infectious Diseases, AIDS and Clinical Immunology Center, 16. Al Kazbegi Ave, 0102 Tblisi, Georgia; rukhadze.nino@yahoo.com; 13Infectious Disease Service, University Hospital Center of Tirana, Rr. Didres, Nr 372, 33979 Tirana, Albania; harxhiarjan@yahoo.com; 14National Institute of Hematology and Infectious Diseases, South-Pest Central Hospital, National Center of HIV, 1097 Budapest, Hungary; btlakatos@gmail.com; 15Faculty of Medicine in Plzeň, University Hospital Plzeň, Charles University, 30599 Plzen, Czech Republic; sedlacek@fnplzen.cz; 16Department of Pharmacology, Clinical Pharmacology and Toxicology, School of Medicine; University of Belgrade, Dr Subotica 1/III, 24101 Belgrade, Serbia; gozza@beotel.net; 17Astar Medical Center, 79041 Lviv, Ukraine; vasylyevmarta@gmail.com; 18Department for Infectious Diseases, Faculty of Medicine, University of Banja Luka, Republika Srpska, 78000 Banja Luka, Bosnia and Herzegovina; antonija@blic.net; 19Department for AIDS, Specialized Hospital for Active Treatment of Infectious and Parasitic Disease Sofia, Medical University of Sofia, 1233 Sofia, Bulgaria; nyancheva@gmail.com

**Keywords:** HIV, COVID-19, ECEE, pneumonia, ARDS, SARS-CoV-2

## Abstract

HIV-positive patients may present lungs with multiple infections, which may hinder differential diagnoses and the choice of treatment in the course of COVID-19, especially in countries with limited access to high-standard healthcare. Here, we aim to investigate the association between radiological changes and poor COVID-19 outcomes among HIV-positive patients from Central and Eastern Europe. Between November 2020 and May 2021, the Euroguidelines in Central and Eastern Europe Network Group started collecting observational data on HIV and COVID-19 co-infections. In total, 16 countries from Central and Eastern European submitted data (eCRF) on 557 HIV-positive patients. The current analyses included patients who had a radiological examination performed. Logistic regression models were used to identify the factors associated with death, ICU admission, and partial recovery (poor COVID-19 outcomes). Factors that were significant in the univariate models (*p* < 0.1) were included in the multivariate model. Radiological data were available for 224 (40.2%) patients, 108 (48.2%) had computed tomography, and 116 (51.8%) had a chest X-ray. Of these, 211 (94.2%) were diagnosed using RT-PCR tests, 212 (94.6%) were symptomatic, 123 (55.6%) were hospitalized, 37 (16.6%) required oxygen therapy, and 28 (13.1%) either died, were admitted to ICU, or only partially recovered. From the radiologist’s description, 138 (61.6%) patients had typical radiological changes, 18 (8.0%) atypical changes, and 68 (30.4%) no changes. In the univariate models, CD4 count (OR = 0.86 [95% CI: 0.76–0.98]), having a comorbidity (2.33 [1.43–3.80]), HCV and/or HBV co-infection (3.17 [1.32–7.60]), being currently employed (0.31 [0.13–0.70]), being on antiretroviral therapy (0.22 [0.08–0.63]), and having typical (3.90 [1.12–13.65]) or atypical (10.8 [2.23–52.5]) radiological changes were all significantly associated with poor COVID-19 outcomes. In the multivariate model, being on antiretroviral therapy (OR = 0.20 [95% CI:0.05–0.80]) decreased the odds of poor COVID-19 outcomes, while having a comorbidity (2.12 [1.20–3.72]) or either typical (4.23 [1.05–17.0]) or atypical (6.39 [1.03–39.7]) radiological changes (vs. no changes) increased the odds of poor COVID-19 outcomes. Among HIV patients diagnosed with symptomatic SARS-CoV-2 infection, the presence of either typical or atypical radiological COVID-19 changes independently predicted poorer outcomes.

## 1. Introduction

The course of the Coronavirus 2019 disease (COVID-19), caused by SARS-CoV-2, can range from the asymptomatic, through mild to critical clinical presentations. The most common of these are respiratory symptoms, and the course of disease follows the fact that the virus multiplies and establishes itself in the respiratory tract where localized inflammation follows [1]. Infiltrates in the lungs are typically seen in chest X-rays (CXR) or computed tomography (CT).

Bilateral consolidations that have a tendency toward the lungs’ periphery are usually found in CXRs and have an appearance that is most consistent with viral pneumonia. Chest CT images are most notable for showing bilateral and peripheral ground glass and consolidated opacities and are marked by an absence of concomitant pulmonary nodules, cavitation, adenopathy, or pleural effusions. These changes are considered typical for COVID-19 and may predict clinical deterioration [2] (Figure 1A–C). The rapid recognition of which stage the patient is at and the deployment of the appropriate therapy will have the greatest benefit [1]. However, it is likely that immunocompromised patients, such as those who are HIV-positive, will have the same presentation for opportunistic infections such as *P. jirovecii*. In addition, they may present with many other types of opportunistic lung infection, such as mycobacteriosis diseases, which may hinder differential diagnosis and the choice of treatment during the course of COVID-19. As a result, patients co-infected with SARS-CoV-2 and HIV might be at higher risk of unfavorable outcomes.

Moreover, it is well established that older age and certain comorbidities, such as diabetes or cardiovascular diseases, may contribute to poorer COVID-19 outcomes in the general population [3]. HIV-positive patients are more likely to develop noncommunicable diseases, to have them occurring earlier in life, and to develop multimorbidity [4,5].

As a result, HIV-positive patients are both prone to lung infections and have a much higher incidence of comorbidities potentially worsening the course of COVID-19.

Therefore, the aim of our study was to analyze radiological changes and their association with COVID-19 outcomes among HIV-positive patients from Central and Eastern Europe.

## 2. Materials and Methods

From November 2020 to May 2021, Euroguidelines in Central and Eastern Europe Network Group collected observational data on HIV and COVID-19 co-infected patients. In total, 16 countries from Central and Eastern European submitted data on 557 HIV-positive patients using an electronic case report form (eCRF) built on the SurveyMonkey^®^ platform. Data was collected from Poland, the Czech Republic, Ukraine, Croatia, Turkey, Romania, Belarus, Estonia, Lithuania, Greece, Georgia, Albania, Hungary, Serbia, Bosnia and Herzegovina, and Bulgaria. Clinical data included demographics, lifestyle, HIV viral load (VL), Lymphocyte CD4+ cell count, history of antiretroviral treatment, and COVID-19 clinical course. Non-AIDS-related comorbidities such as cardiovascular, respiratory disease, kidney disease, diabetes, and malignancy were also included. All patients were under the care of specialized HIV outpatient clinics. Patients’ outcomes were reported at the time of completion of the survey as full recovery, partial recovery, death, currently still in hospital, or unknown. In addition, information about hospitalization (yes/no) and ICU admittance (yes/no) was requested.

The current analyses included patients who had a radiological examination performed (*n* = 224) and patients with a known outcome (*n* = 214). In the analysis, only results from the first performed radiological imaging were included. The selection of patients who were eligible for analyses was described according to the STROBE protocol [6]. COVID-19 diagnosis was predominately based on positive swabs tested using reverse transcriptase polymerase chain reaction (RT-PCR), while some cases were confirmed using the serology assessment of IgM and IgG antibody titers; in a few cases, the diagnosis was based on radiological imaging. Study outcome was defined as a composite outcome with death, intensive care unit (ICU) admission, or partial recovery (poor COVID-19 outcome). Patients with an unknown outcome, still in hospital at the time of data collection, or who were diagnosed with HIV while being infected with SARS-CoV-2 were excluded from the analysis.

In the statistical analyses, non-parametric tests were used as appropriate for group comparisons.

Logistic regression models were used to identify factors associated with death, intensive care unit (ICU) admission, and no improvement (poor COVID-19 outcomes). Factors that were significant in the univariate models (*p* < 0.1) were included in the multivariate model. We created one model for both typical and atypical changes versus no radiological changes, and another with these groups combined into “any radiological changes” compared to no changes. Statistical analysis was performed using SAS version 9.4 (SAS Institute, Cary, NC).

The design of this work conforms to the standards currently applied in the Medical University of Warsaw’s Bioethics Committee. Approval number: AKBE/155/2020.

## 3. Results

Radiological data were available for 224/557 (40.2%) patients: 108/224 (48.2%) had computed tomography, and 116/224 (51.8%) had a chest X-ray. Of these, 211/224 (94.2%) were diagnosed using the RT-PCR test, 212/224 (94.6%) were symptomatic, 123/224 (55.6%) were hospitalized, 37/224 (16.6%) required oxygen therapy, and 28/224 (13.1%) either died, were admitted to ICU, or only partially recovered. From the radiologist’s description, 138/224 (61.6%) patients had typical radiological changes, 18/224 (8.0%) had atypical changes, and 68/224 (30.4%) had no changes (group characteristics are presented in Table 1).

Patients with typical or atypical radiological changes were older than the patients without any changes (47 years [IQR: 38.5–57] vs. 45.5 years [IQR: 38–52] vs. 40 years [IQR: 34.5–48.5], *p* = 0.008, respectively). Moreover, patients with atypical radiological changes had a lower BMI compared to patients with typical or no radiological changes (20.8 [IQR: 17.8–24.4] vs. 24.6 [IQR: 21.6–29.0] vs. 24.0 [IQR: 21.3–29.0], *p* = 0.0044, respectively). Patients with atypical radiological changes were more likely to have HCV and/or HBV co-infections compared to patients with typical or no radiological changes (44.4% vs. 17.5% vs. 10.8%, *p* = 0.0017, respectively). Patients with atypical changes were diagnosed with HIV earlier compared to patients with typical or no changes (11.5 years [IQR: 1–19] vs. 10 years [IQR: 6–15] vs. 7 years [IQR 3–11], *p* = 0.0107, respectively). Patients with no radiological changes were more likely to have undetectable HIV VL compared to patients with typical or atypical changes (82.3% vs. 62.6% vs. 50%, *p* = 0.0114, respectively). However, patients with typical changes were more commonly undergoing cART treatment than patients with atypical or no changes (94.2% vs. 72.2% vs. 88.2%, *p* = 0.078, respectively). Patients with typical changes were more likely to have COVID-19 symptoms compared to patients with atypical or no changes (97.8% vs. 94.4% vs. 88.2%, *p* = 0.0160, respectively). Patients with atypical changes were more likely to be hospitalized due to COVID-19 compared to patients with typical or no radiological changes (72.2% vs. 63.8% vs. 32.8%, *p* < 0.0001, respectively). However, patients with typical radiological changes more often required oxygen therapy in comparison to patients with atypical or no radiological changes (91.9% vs. 8.1% vs. 0%, *p* < 0.0001, respectively). Patients without any radiological changes were less likely to die, be admitted to ICU, or to have no clinical improvement compared to patients with typical or atypical changes (4.4% vs. 15.3% vs. 33.3%, *p* = 0.0054, respectively).

Patients who had a poor COVID-19 outcome were more likely to have one or more comorbidities (57.1% vs. 32.8%, *p* = 0.0190), HCV and/or HBV co-infections (35.7% vs. 14.8%, *p* = 0.0228), were more often hospitalized (96.4% vs. 47.6%, *p* < 0.001), more often required oxygen therapy (46.4% vs. 11.9%, *p* < 0.001), and more often had radiological changes (89.3% vs. 65.1%, *p* < 0.001). Patients who had a poor COVID-19 outcome were less often on cART (75% vs. 93%, *p* = 0.0073) and had a lower median CD4+ cell count (403 [IQR: 172–582.5] vs. 568 [IQR: 348–861], *p* = 0.0099).

In univariate models, testing all variables in Table 2, the following variables were significantly associated with poor COVID-19 outcomes: CD4+ count (OR = 0.96 [95% CI: 0.8–1.1, *p* = 0.0225]), having a comorbidity (2.33 [95% CI: 1.43–3.80, *p* = 0.0007]), having HCV and/or HBV co-infections (3.17 [95% CI: 1.32–7.60], *p* = 0.0097), currently employed (0.31 [95% CI: 0.13–0.70], *p* = 0.0051), being on antiretroviral therapy (0.22 [95% CI: 0.08–0.63, *p* = 0.0044]), and having typical (3.90 [95% CI: 1.12–13.65], *p* = 0.0124) or atypical (10.8 [95% CI: 2.23–52.5], *p* = 0.0124) radiological changes, but also having no radiological changes (4.48 [95% CI: 1.3–15.39], *p* = 0.0174).

In the multivariate model, being on antiretroviral therapy (OR = 0.20 [95% CI:0.05-0.80], *p* = 0.0231) decreased the odds of poor COVID-19 outcomes. Having a comorbidity (2.12 [1.2–3.7], *p* = 0.0091), or either typical (4.23 [95% CI: 1.05–17.0], *p* = 0.0418) or atypical (6.39 [95% CI: 1.03–39.7], *p* = 0.0465) radiological changes increased the odds of poor COVID-19 outcomes (Table 3). In addition, we built a model that included all the above-mentioned variables, but that combined typical and atypical radiological changes into one category, which was then compared to no radiological changes. In this model, the OR for any radiological changes was 4.57 [95% CI: 1.16–18], *p* = 0.03), while all other effects remained within the same trends (data not shown).

## 4. Discussion

In our study, we found that the presence of any radiological changes increased the odds of death, ICU admission, or partial recovery by almost five times. When the data was stratified by the type of changes, the effect was still independent and significant. To the best of our knowledge, this is the first study on HIV-positive patients with the COVID-19 disease in which radiological imaging was analyzed for its impact on prognosis. Other studies in this area were more focused on the prevalence of radiological changes than their impact on patient outcome.

Gurumurthy et al. [7] investigated 298 confirmed COVID-19 cases who underwent a chest CT. Typical features were found in 218/298 (73.1%) cases, atypical radiological changes were present in 63/218 (21.1%) cases, and no changes were found in the remaining 17/298 (5.8%) cases. A significant, positive correlation between the CT severity score for the atypical group and age was observed in their study (*ρ* = 0.343 and *p* = 0.006), indicating that with increasing age there was an increase in the CT severity score and also atypical CT features. However, there was no statistically significant correlation between typical or atypical CT features and the severity of the disease (CT severity score). No statistically significant association between typical or atypical CT features and mortality was noted either [7]. In our study, we investigated patients with a concomitant HIV infection. Moreover, we have shown that the occurrence of any radiological change is an independent risk factor for poor COVID-19 outcomes.

*Pneumocystis jirovecii* pneumonia (PCP) is a common opportunistic fungal infection that affects immunosuppressed patients including HIV-positive patients. The most common high-resolution CT finding for PCP is a diffuse ground-glass opacity. Consolidation, nodules, cysts, and spontaneous pneumothorax can also develop. These changes, if they occur in the course of COVID-19, may pose a diagnostic challenge, especially when HIV status is unknown [8]. Therefore, as already recommended by the Polish AIDS Clinical Society [9], patients with a COVID-19 infection and radiological changes should be tested for HIV. All patients with a confirmed HIV infection should undergo a differential diagnosis including PCP, which is crucial for administering proper etiotropic treatment. To date, there have only been case reports published on concomitant HIV, COVID-19, and PCP in this area [10,11]. Chong et al. [12], in their review of the relationship between COVID-19 and PCP, identified 12 cases with concomitant diseases and concluded that there is huge variability in the timing from illness onset to presentation, and presentation to PCP diagnosis in COVID-19 patients, regardless of HIV status. Whether *Pneumocystis jirovecii* colonizes the lungs during COVID-19 or is a contributing factor along with SARS-CoV-2 remains, in some cases, unclear. It is essential to maintain a broad differential diagnosis, and prudent to consider additional diagnostic testing for *P. jirovecii* in COVID-19 patients, especially when there is a lack of clinical improvement in respiratory status, radiological features indicating lung cysts and, possibly, pneumothorax; and laboratory findings of elevated serum lactate dehydrogenase (LDH) and beta-d-glucan (BDG) levels, even in the absence of classical risk factors such as HIV [12].

Another independent risk factor for poor COVID-19 outcomes that was found in our study was having a non-HIV-related comorbidity, which increased the odds of poor outcomes by over two times. At the same time, being on cART decreased these odds by 80%.

Gerreti et al. [13], in their study, compared 47,702 patients who had COVID-19 to 122 HIV/COVID-19 coinfected patients, and found evidence suggesting that age, sex, and HIV infection all affect the risk of ICU admission. Moreover, sex, ethnicity, age, baseline date, indeterminate/probable hospital acquisition of COVID-19, presence of comorbidities, hypoxia, and/or oxygen requirement on admission affected risk of death due to COVID-19. In our study, comorbidities including HBV and/or HCV co-infections, as well as oxygen therapy, were independently associated with poorer outcomes in univariate regression but were no longer significant after adjustment. However, our cohort consisted only of HIV-positive, SARS-CoV-2 coinfected patients, whereas Gerreti et al. compared HIV-positive patients with HIV-negative patients. In addition, they did not focus on such factors as the presence of radiological changes in imaging testing, being on cART, or lymphocyte CD4+ count, which we found affected the outcome of COVID-19 [13].

Sarkar et al., in their systematic review and meta-analysis of the impact of SARS-CoV-2 infected patients with concurrent co-infections, concluded that the clinical outcomes of COVID-19 in HIV-positive patients or those with chronic hepatitis are comparable to COVID-19 patients without these concurrent infections [14].

Hadi et al. [15] enrolled 49,763 COVID-19 patients and compared them with 404 HIV/COVID-19 coinfected patients. The propensity-matched analyses revealed no difference in outcomes, showing that higher mortality was driven by a higher burden of comorbidities [15]. These findings are in line with our study, showing that, for HIV-positive patients, the major risks are from a burden of non-HIV-related co-morbidities.

Due to the fact that our study is of a retrospective observational nature some important limitations are present. First, our study is likely to underestimate the rate of SARS-CoV-2 infection among HIV-positive patients by the number of patients not seeking medical care due to mild or asymptomatic course of disease. In addition, patients were selected for analyses because they had a radiological examination performed. Both of these factors could overestimate the risk of poor outcomes in patients with HIV and COVID-19. However, as discussed above, the mortality rate in our study is comparable to other studies **[13,15]**. Second, 90.6% of our patients were on cART and had a median CD4+ count of 539 cells/uL (IQR: 307–818 cells/uL), which indicates that the majority of our cohort was retained in care with well-controlled HIV. Thus, we have limited power to analyze the outcomes of patients with advanced HIV, or who are not on cART. However, there are also some strengths worth mentioning: the eCRF design included information such as time since HIV diagnosis, the patient’s immunological status, mode of HIV infection, and their most recent cART regimen. Moreover, in order to collect these data, as well as to allow for follow-up after the patient’s discharge, we approached these patients’ physicians in specialist HIV care. This allowed us to enlarge our cohort with patients who were not hospitalized and who could be followed at home or in HIV outpatient clinics.

## 5. Conclusions

Among HIV patients diagnosed with a symptomatic SARS-CoV-2 infection, the presence of either typical or atypical radiological COVID-19 changes independently predicted poorer outcomes. Due to the overlap in the clinical and radiological presentation of some opportunistic infections and COVID-19, identifying HIV status is crucial for further treatment outcomes. Moreover, non-HIV-related concomitant comorbidities and not being on cART were independent risk factors for poor outcomes for COVID-19.

## Figures and Tables

**Figure 1 viruses-14-00972-f001:**
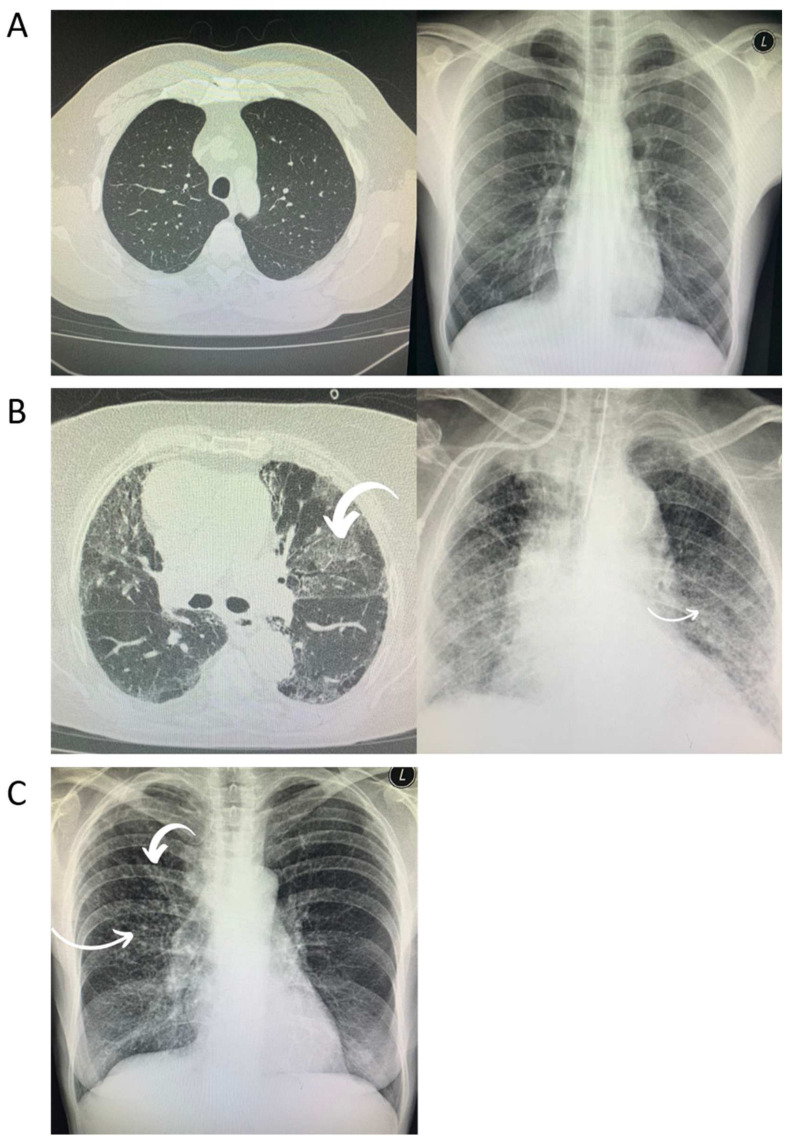
Radiological chest imaging of HIV-positive patients with COVID-19. (**A**) These images depict no radiological changes. (**B**) Typical radiological changes (bilateral and peripheral ground glass and consolidated opacities). (**C**) Atypical radiological changes (diffuse nodular changes).

**Table 1 viruses-14-00972-t001:** Baseline characteristics of HIV/COVID-19 coinfected patients relating to the occurrence of radiological changes in chest X-rays or computed tomography.

Characteristic		All *n* = 224	Radiological Changes			*p*-Value
			Typical *n* = 146	Atypical *n* = 22	No Changes *n* = 70	
Demographics	**Age in years, median** **(IQR)**	**45 (35.0–55.0)**	**47.0 (38.5–57.0)**	**45.5 (38.0–52.0)**	**40 (34.5–48.5)**	**0.0080**
	**BMI in kg/m^2^, median** **(IQR)**	**24.6 (21.4–28.7)**	**24.6 (21.6–29.0)**	**20.9 (17.8–24.4)**	**24.0 (21.3–29.0)**	**0.0044**
	Female sex, *n* (%)	77 (34.7)	44 (32.1)	7 (41.2)	26 (38.2)	0.5788
	Currently employed, *n* (%)	133 (59.4)	82 (59.4)	10 (55.6)	41 (55.6)	0.9358
Comorbidities	Always smoked cigarettes, *n* (%)	136 (60.7)	54 (61.4)	12 (66.7)	40 (58.8)	0.8308
	One or more comorbidities, *n* (%)	83 (37.0)	53 (38.4)	9 (50.0)	21 (30.9)	0.2848
	Number of comorbidities, median (IQR)	0 (0–1)	0 (0–1)	1 (0–1)	0 (0–1)	0.2281
	**HCV and/or HBV co-infection (%)**	**39 (17.7)**	**24 (17.5)**	**8 (44.4)**	**7 (10.8)**	**0.0017**
	mode of HIV infection					
HIV characteristics	MSM, *n* (%)	64 (28.6)	40 (29.0)	5 (27.8)	19 (27.9)	0.1342
	Heterosexual, *n* (%)	101 (45.1)	62 (44.9)	8 (44.4)	31 (45.6)	
	IDU, *n* (%)	43 (19.2)	30 (21.7)	4 (22.2)	9 (13.2)	
	Other, *n* (%)	16 (7.1)	6 (4.3)	1 (5.6)	9 (13.2)	
	**Time since HIV diagnosis in years, median (IQR)**	**9 (5–14)**	**10 (6–15)**	**11.5 (1–19)**	**7 (3–11)**	**0.0107**
	CD4 count in cells/uL, median (IQR)	539 (307–818)	545 (370–830)	344 (140–609)	521 (268–833)	0.1017
	**HIV VL < 50 copies/mL, *n* (%)**	**174 (77.7)**	**109 (62.6)**	**9 (50.0)**	**56 (82.3)**	**0.0114**
	**On cART, *n* (%)**	**203 (90.6)**	**130 (94.2)**	**13 (72.2)**	**60 (88.2)**	**0.0078**
	InSTI as third drug in cART, *n* (%)	134 (65.4)	81 (62.3)	11 (73.3)	42 (70.0)	0.2546
	TDF or TAF in backbone, *n* (%)	146 (65.2)	90 (65.2)	13 (72.2)	43 (63.2)	0.7762
**COVID-19 characteristics**	**Any COVID-19 symptoms, *n* (%)**	**212 (94.6)**	**135 (97.8)**	**17 (94.4)**	**60 (88.2)**	**0.0160**
	**Hospitalized, *n* (%)**	**123 (55.6)**	**88 (63.8)**	**13 (72.2)**	**22 (32.8)**	**<0.0001**
	**Requiring oxygen therapy, *n* (%)**	**37 (16.6)**	**34 (91.9)**	**3 (8.1)**	**0 (0)**	**<0.0001**
	**Died, admitted to ICU, or no improvement, *n* (%)**	**28 (13.1)**	**20 (15.3)**	**5 (33.3)**	**3 (4.4)**	**0.0054**

**Table 2 viruses-14-00972-t002:** Baseline characteristics of HIV/COVID-19 co-infected patients, stratified by COVID-19 outcome.

Characteristic		All *n* = 214	COVID-19 Outcome		*p*-Value
			Full Recovery *n* = 186	Death/ICU or Partial Recovery *n* = 28	
Demographics	Age in years, median (IQR)	45 (37–55)	45 (37–55)	44 (44–57)	0.6103
	BMI in kg/m^2^, median (IQR)	24.6 (21.4–28.7)	24.8 (22–29)	23.1 (19–29.2)	0.2379
	Female sex, *n* (%)	75 (35.4)	64 (34.8)	11 (39.3)	0.6743
	Currently employed, *n* (%)	130 (60.8)	120 (64.5)	10 (35.7)	0.9990
Comorbidities	Always smoked cigarettes, *n* (%)	129 (60.3)	110 (59.1)	19 (67.9)	0.4152
	**One or more comorbidities, *n* (%)**	**77** **(36)**	**61** **(32.8)**	**16** **(57.1)**	**0.0190**
	**Number of comorbidities, median (IQR)**	**0** **(0–1)**	**0** **(0–1)**	**1** **(0–1)**	**0.0043**
	**HCV and/or HBV co-infection, *n* (%)**	**37** **(17.54)**	**27** **(14.8)**	**10** **(35.7)**	**0.0228**
HIV characteristics	Mode of HIV infection				
	MSM, *n* (%)	62 (29)	52 (28)	10 (35.7)	0.8756
	Heterosexual, *n* (%)	99 (46.3)	88 (47.3)	11 (39.3)	
	IDU, *n* (%)	37 (17.3)	32 (17.2)	5 (17.9)	
	Other, *n* (%)	16 (7.5)	14 (7.5)	2 (7.4)	
	Time since HIV diagnosis in years, median (IQR)	9 (5–14)	9 (5–14)	8.5 (3.5–14.5)	0.6448
	**CD4 count in cells/uL, median (IQR)**	**539.5 (307–818)**	**568** **(348–861)**	**403** **(172–582.5)**	**0.0099**
	HIV VL < 50 copies/mL, *n* (%)	168 (78.5)	149 (80.1)	19 (67.9)	0.1455
	**On cART, *n* (%)**	**194** **(90.7)**	**173** **(93)**	**21** **(75)**	**0.0073**
	InSTI as third drug in cART, *n* (%)	127 (65.1)	114 (65.9)	13 (59.9)	0.2106
	TDF or TAF in backbone, *n* (%)	139 (65)	124 (66.7)	15 (53.6)	0.2042
**COVID-19 characteristics**	Any COVID-19 symptoms, *n* (%)	202 (94.4)	176 (94.6)	26 (92.9)	0.6600
	**Hospitalized, *n* (%)**	**115** **(54)**	**88** **(47.6)**	**27** **(96.4)**	**<0.0001**
	**Requiring oxygen therapy, *n* (%)**	**35** **(16.4)**	**22** **(11.9)**	**13** **(46.4)**	**<0.0001**
	**No radiological changes, *n* (%)**	**68** **(31.8)**	**65** **(34.9)**	**3** **(10.7)**	**0.0054**
	**non-typical radiological changes, *n* (%)**	**15** **(7)**	**10** **(5.4)**	**5** **(17.9)**	
	**typical radiological changes, *n* (%)**	**131** **(61.2)**	**111** **(59.7)**	**20** **(71.4)**	
	**any radiological changes, *n* (%)**	**146** **(68.2)**	**121** **(65.1)**	**25** **(89.3)**	**0.0090**

**Table 3 viruses-14-00972-t003:** Univariate and multivariate logistic regression analyses of the factors associated with poor COVID-19 outcome in HIV-positive patients.

Factor		Univariate			Multivariate		
		Odds Ratio	95% Confidence Interval	*p*-Value	Odds Ratio	95% Confidence Interval	*p*-Value
Demographics	Age [unit = 10]	1.17	0.85–1.62	0.3296	-	-	-
	BMI [unit = 1]	0.98	0.91–1.06	0.6678	-	-	-
	Male sex	0.82	0.36–1.87	0.6428	-	-	-
	Currently employed	0.31	0.13–0.7	0.0051	0.46	0.18–1.22	0.1201
Comorbidities	Always smoked cigarettes	1.46	0.63–3.97	0.3816	-	-	-
	**One or more comorbidities**	**2.73**	**1.22–6.13**	**0.0148**	**2.52**	**1–6.3**	**0.0490**
	**Number of comorbidities** **[unit = 1]**	**2.33**	**1.43–3.8**	**0.0007**	**2.12**	**1.2–3.7**	**0.0091**
	HCV and/or HBV co-infection	3.17	1.32–7.60	0.0097	1.51	0.5–4.5	0.4636
	mode of HIV infection						
HIV characteristics	heterosexual vs. MSM	0.65	0.26–1.64	0.8786	-	-	-
	IDU vs. MSM	0.81	0.26–2.6		-	-	-
	Other vs. MSM	1.3	0.131–12.88		-	-	-
	Unknown vs. MSM	0.52	0.06–4.53		-	-	-
	Time since HIV diagnosis in years [unit = 1]	1	0.94–1.06	0.9932	-	-	-
	CD4 count [unit = 100]	0.86	0.8–1	0.0225	0.92	0.8–1.1	0.2134
	HIV VL > = 50 copies/mL	1.91	0.8–4.56	0.1461	-	-	-
	**On cART**	**0.23**	**0.08–0.63**	**0.0044**	**0.2**	**0.05–0.8**	**0.0231**
	Third drug in cART						
	InSTI vs. PI	0.46	0.17–1.25	0.2523	-	-	-
	NNRTI vs. PI	0.15	0.02–1.34		-	-	-
	Other vs. PI	0.8	0.08–7.99		-	-	-
	No TDF or TAF in backbone	1.73	0.78–3.87	0.1793	-	-	-
**COVID-19 characteristics**	Any COVID-19 symptoms	0.74	0.15–3.56	0.7055	-	-	-
	**typical radiological changes**	**3.9**	**1.12–13.65**	**0.0124**	**4.23**	**1.06–16.99**	**0.0418**
	**non-typical radiological changes**	**10.8**	**2.23–52.5**	**0.0124**	**6.39**	**1.02–39.72**	**0.0465**

## Data Availability

The data sets used and/or analyzed during the current study can be made available by the corresponding author on reasonable request.

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
