# Peer review of "The Presence of Either Typical or Atypical Radiological Changes Predicts Poor COVID-19 Outcomes in HIV-Positive Patients from a Multinational Observational Study: Data from Euroguidelines in Central and Eastern Europe Network Group"

_viruses, 2022, doi:10.3390/v14050972_

Round 1
Reviewer 1 Report
This is a New manuscript submission from Kowalska et al at Medical University of Warsaw in Polland and multi-center collaborators in Central and Eastern Europe.
The submission focuses on documenting the outcomes of HIV-infected patients with COVID-19 based on radiological changes, which is very significant in the field because we are in need of collective documentation to understand how COVID impacted HIV patients as susceptible population. The authors report that, of 224 HIV+ patients who had reasons to have a pulmonary evaluation, 61.6% presented typical radiological changes. These were most likely to have COVID symptoms and required oxygen therapy. They also report that 8% of patients had atypical radiological changes and that the features in this case included older patients, longer HIV infection, and generally more hospitalizations due to COVID. The study also describes 30.4% of HIV patients who had no radiological changes and that these patients most likely exhibited undetectable HIV VLs.
The manuscript is generally well written and organized.
Strengths include:
- the relevance of the problem being studied,
- the amount of clinical information collected,
- and the multi-variate analyses approach.
Some weakness include:
- Limitation of the fact that only 224 HIV-infected cases were identified in 16 countries by means of a Survey Monkey platform. This seems to be a low response rate.
- In addition, the study is based only on those HIV patients who had reasons to have a pulmonary evaluation, but this is intrinsic to the fact that this is a retrospective observational study, which the authors acknowledge.
- What exactly was deemed as typical or atypical changes? This must be clearly delineated.
- The manuscript would benefit tremendously from actual photographs serving as examples of typical/atypical/none radiological changes per group.
Minor:
- Tables would benefit from somehow highlighting the significant values (characteristics) either by adding stars or bolding the text. This would make it easier for the reviewer to find the categorical relevance.
- In addition, sorting the data by p-value might help. Alternatively, the authors may choose to itemize the characteristics with subheadings for demographics, viral parameters, immune parameters, lifestyle, co-infections, co-morbidities, as appropriate.
Author Response
Response to Reviewer 1 Comments
Point 1:
This is a New manuscript submission from Kowalska et al at Medical University of Warsaw in Polland and multi-center collaborators in Central and Eastern Europe.
The submission focuses on documenting the outcomes of HIV-infected patients with COVID-19 based on radiological changes, which is very significant in the field because we are in need of collective documentation to understand how COVID impacted HIV patients as susceptible population. The authors report that, of 224 HIV+ patients who had reasons to have a pulmonary evaluation, 61.6% presented typical radiological changes. These were most likely to have COVID symptoms and required oxygen therapy. They also report that 8% of patients had atypical radiological changes and that the features in this case included older patients, longer HIV infection, and generally more hospitalizations due to COVID. The study also describes 30.4% of HIV patients who had no radiological changes and that these patients most likely exhibited undetectable HIV VLs.
The manuscript is generally well written and organized.
Response 1: Thank you for the acknowledgement of our work.
Point 2:
Strengths include:
- the relevance of the problem being studied,
- the amount of clinical information collected,
- and the multi-variate analyses approach.
Response 2: Thank you for the acknowledgement of the strengths of our work.
Point 3:
Some weakness include:
- Limitation of the fact that only 224 HIV-infected cases were identified in 16 countries by means of a Survey Monkey platform. This seems to be a low response rate.
- Thank you for this remark. We have collected data on 557 HIV-positive individuals from 16 HIV centers between November 2020 and May 2021. We could estimate this centers are following up to 8 000 patients, which gives the overall prevalence of COVID of 9.4%. In comparison the overal COVID prevalence in Poland is around 14%. Therefore we agree that we could underestimate the number of COVID infections by 30%, however these patients who were not reporting to HIV centers with COVID infection were most likely to have a mild course of diseases with no radiological examination performed. Of these patients 224 had the radiological examination performed and were included into this analysis
- We have now added this explanation to limitation section.
- In addition, the study is based only on those HIV patients who had reasons to have a pulmonary evaluation, but this is intrinsic to the fact that this is a retrospective observational study, which the authors acknowledge.
- Thank you for this remark. Yes we aimed to evaluate the association between the radiological changes and outcome of COVID-19 in HIV-positive individuals. Therefore, only patients with radiological pulmonary evaluation could have been included in our analysis.
- What exactly was deemed as typical or atypical changes? This must be clearly delineated.
- We agree with the reviewer that there is no standardized definition of “typical” and “atypical” description. After reviewing the available literature (ref) we defined it in the methods section as: “Bilateral consolidations that have a tendency toward the lungs’ periphery are usually found in CXRs, and have an appearance that is most consistent with viral pneumonia. Chest CT images are most notable for showing bilateral and peripheral ground glass and consolidated opacities, and are marked by an absence of concomitant pulmonary nodules, cavitation, adenopathy, or pleural effusions. These changes are considered typical for COVID-19, and may predict clinical deterioration”. Presence of other radiological changes than those defined as typical was considered atypical.
- The manuscript would benefit tremendously from actual photographs serving as examples of typical/atypical/none radiological changes per group.
- Thank you for this constructive idea. We have selected most representative examples from the group of HIV+COVID+ patients hospitalized in our center and added it as figure.
Point 4:
Minor:
- Tables would benefit from somehow highlighting the significant values (characteristics) either by adding stars or bolding the text. This would make it easier for the reviewer to find the categorical relevance.
- Thank you for this remark. We have bolded in every table the significant characteristics.
- In addition, sorting the data by p-value might help. Alternatively, the authors may choose to itemize the characteristics with subheadings for demographics, viral parameters, immune parameters, lifestyle, co-infections, co-morbidities, as appropriate.
- Thank you for this remark. We have itemized the characteristics with subheadings.

Reviewer 2 Report
The authors investigated the association between radiological changes and poor Covid-19 outcomes in a multicenter registry. Their main message is that radiological changes independently predicted poor outcomes.
I am quite asthonished that the authors present such a trivial association as the main message of the analysis of a database of more than 500 documented HIV patients with covid. Covid-19 has a wide spectrum of clinical courses ranging from asymptomatic infection to severe pneumonia. It is obvious that patients with radiological changes documenting pneumonia have a more severe disease and a worse prognosis than patients without radiological changes and thus without pneumonia. I cannot see a novelty in such a trivial message. Therefore, I don´t understand why the authors present their data in such a way. Either they provide new detailed data on specific radiological symptoms beyond extent of affected lung or they should concentrate on risk factors for a more severe course of Covid in HIV infected subjects. As they have data on a large number of patients, it is important to analyze risk factors for a severe course of infection.
In the introduction they write that the aim of the study was to describe radiological changes and their association with outcomes. However, there are no radiological changes described except for grouping into typical and atypical changes. If they want to analyze differences between patients with typical and atypical radiological signs, then they must given a detailed description of the atypical signs and also provide information regarding the extent of changes.
They do not provide data whether the atypical changes were caused by other infections or whether they only present variation of Covid-19 radiological signs. There is no definition for typical and atypical radiological signs in the manuscript. There is no information regarding the extent of radiological signs. If the focus of the manuscript is on radiological signs, then the presentation is inadequate.
In the Materials section it is stated that the selection of patients was performed according to the STROBE Protocol. This is strange as the STROBE protocol focuses on the presentation of observational studies but to my knowledge it does not address selection processes. The patients don´t provide any information regarding the selection of patients. What was the criterium for entry into the hospital? What were the criteria of the selection and what is the influence of the STROBE protocol on the selection process? The authors should use the primary references that published the STROBE protocol for the first time and not secondary references.
The authors write: “The current analyses included patients who had a radiological examination performed, and patients with a known outcome”. It is not documented at which time point the radiological examination was performed and if consecutive examinations were performed, which result was included? The first examination or the examination with the worst change? A correlation to outcome only makes sense when only radiological examinations at hospital entry would have been included. As ICU was an endpoint: were radiological examinations at ICU wards excluded?
The authors combine for the assessment of poor Covid-19 outcomes three factors: death, ICU admission, no improvement. The factor no improvement makes sense only if a time span is given. There is no indication of the corresponding time period in which improvement was expected. There is no indication what “no improvement” means: ongoing artificial ventilation, ICU treatment or a persistent sore throat? Data for the individual factors must be given and also the analyses should be done in the individual groups. What are risk factors for mortality or treatment at the ICU ? Number of deaths and causes of deaths should be given,The entry criteria for ICU are quite different in hospitals. In some hospitals everyone on high flow oxygen is admitted to ICU in others only if the are artificially ventilated. The criteria should be described.
For analyzing the role of radiological signs to death, it is important to document the causes of death such as ARDS, myocardial infection, pulmonary embolism, septic shock.
They report that patients with hepatitis C and B infection have more atypical radiological signs. As these viruses don´t produce radiological signs except for very rare autoimmune phenomena, these changes should be described: bacterial pneumonia?
In summary, I think it is quite disappointing that the authors have an interesting large data base of HIV-infected patients with Covid and they are not able or not willing to present their data in a way that they can address the important questions. They present a quite trivial conclusion that a SARS-CoV-2-infected patient with a pneumonia has a worse prognosis in comparison to patients without a pneumonia. Instead, they should evaluate their cohort regarding the risk factors for a more severe course of infection, for a worse prognosis, for treatment on the ICU and for death. If atypical radiological signs are caused by other infections, the role of coinfections should be addressed. If specific radiological signs (and not only the extent of lung involvement) are linked to a worse prognosis, then, this should be reported, but the atypical radiological signs must be presented in detail because a wide range of radiological signs are considered atypical.
Radiological signs are a symptom of a more severe Covid-19. If the risk factors for a more severe Covid-19 are evaluated, then it is important to know when the radiological examination occurred in the course of the stay in the hospital and which of consecutive examination was included in the analysis.
If patients with hepatitis B/C have a more severe course of infection, then it should be investigated whether this is caused by hepatitis B/C infection itself or by hepatic insufficiency or cirrhosis.
Author Response
Response to Reviewer 2 Comments
Point 1:
The authors investigated the association between radiological changes and poor Covid-19 outcomes in a multicenter registry. Their main message is that radiological changes independently predicted poor outcomes.
I am quite asthonished that the authors present such a trivial association as the main message of the analysis of a database of more than 500 documented HIV patients with covid. Covid-19 has a wide spectrum of clinical courses ranging from asymptomatic infection to severe pneumonia. It is obvious that patients with radiological changes documenting pneumonia have a more severe disease and a worse prognosis than patients without radiological changes and thus without pneumonia. I cannot see a novelty in such a trivial message. Therefore, I don´t understand why the authors present their data in such a way. Either they provide new detailed data on specific radiological symptoms beyond extent of affected lung or they should concentrate on risk factors for a more severe course of Covid in HIV infected subjects. As they have data on a large number of patients, it is important to analyze risk factors for a severe course of infection.
Response 1: We would like to thank the reviewer for underlining the need for better clarification of the purpose of our study. We agree that in general population the intensity of radiological changes of COVID-19 pneumonia is straight forward related to clinical course of diseases and its outcomes. However this is not the case for HIV positive patients. Radiological presentations in HIV positive patients may include atypical changes representing viral (CMV, VZV, HSV), fungal (candidiasis, aspergillosis), bacterial (tuberculosis, MAC) and parasitic (P. jiroveci) infections. This picture can overlap with changes present in the COVID-19 pneumonia, making the diagnosis difficult and therefore treatment not accurate for the pathogen. This in turn can cause an additional risk to the patient worsening COVID outcomes.
Hence we believe, our results may be beneficial for better understanding the relation between radiological changes and COVID-19 course in HIV-positive individuals.
Point 2:
In the introduction they write that the aim of the study was to describe radiological changes and their association with outcomes. However, there are no radiological changes described except for grouping into typical and atypical changes. If they want to analyze differences between patients with typical and atypical radiological signs, then they must given a detailed description of the atypical signs and also provide information regarding the extent of changes.
Response 2: We agree with reviewer, in fact the most desirable approach would be to collect actual radiological assessments and to have a review committee analyzing it and provided an agreed statement according to predefined categorization. However our study is a real world data analyses, with obvious limitations explained in our manuscript. In addition we were not collecting the original radiological visual records or descriptions and our survey was constructed in the way to standardize the responses into typical or atypical category according to a definition provided in methods section. Having said that we hope that reviewer will acknowledge the fact that we collected the data from 16 CEE countries with modest funds and we were not be able to translate all descriptions provided in multiple languages. However we have added examples of radiological changes as a figure.
Point 3:
They do not provide data whether the atypical changes were caused by other infections or whether they only present variation of Covid-19 radiological signs. There is no definition for typical and atypical radiological signs in the manuscript. There is no information regarding the extent of radiological signs. If the focus of the manuscript is on radiological signs, then the presentation is inadequate.
Response 3: Thank you for this remark. Unfortunately we cannot provide data whether the atypical changes were caused by other infections or whether they only present variation of COVID-19 because we do not have such data; we have collected data on comorbidities including viral coinfections, and presence of radiological changes, but we do not have the microbiological confirmation of the exact etiological factor from lung samples. Moreover from our clinical experience in many of COVID-19 and HIV cases etiology remains unknown even with multiple microbiological tests performed. It is a well established practice to treat tuberculosis basing on radiological picture even in lack of microbiological confirmation. The same practice applies to toxoplasmosis and pneumocystis pneumonia. In addition we would like to point out that in the time when data were collected (November 2020 to May 2021) high incidence of SARS-CoV-2 infections and high mortality was observed limiting the ability to perform in depth diagnostics.
Point 4:
In the Materials section it is stated that the selection of patients was performed according to the STROBE Protocol. This is strange as the STROBE protocol focuses on the presentation of observational studies but to my knowledge it does not address selection processes. The patients don´t provide any information regarding the selection of patients. What was the criterium for entry into the hospital? What were the criteria of the selection and what is the influence of the STROBE protocol on the selection process? The authors should use the primary references that published the STROBE protocol for the first time and not secondary references.
Response 4: Thank you for this remark. We have used the STROBE Protocol to describe the the eligibility criteria for statistical analyses, not for the study itself. As stated we have collected observational data on HIV-positive individuals with COVID-19 into the case report form from 16 countries. For current study analyses we have selected patients with known outcome and who had radiological imaging performed. As for the STROBE reference we have changed it accordingly to what is stated on the STROBE website: https://www.strobe-statement.org/strobe-publications/.
Point 5:
The authors write: “The current analyses included patients who had a radiological examination performed, and patients with a known outcome”. It is not documented at which time point the radiological examination was performed and if consecutive examinations were performed, which result was included? The first examination or the examination with the worst change? A correlation to outcome only makes sense when only radiological examinations at hospital entry would have been included. As ICU was an endpoint: were radiological examinations at ICU wards excluded?
Response 5: Thank you for this remark. We have clarified now this statement in the methods.
Point 6:
The authors combine for the assessment of poor Covid-19 outcomes three factors: death, ICU admission, no improvement. The factor no improvement makes sense only if a time span is given. There is no indication of the corresponding time period in which improvement was expected. There is no indication what “no improvement” means: ongoing artificial ventilation, ICU treatment or a persistent sore throat?
Response 6: All outcomes were assessed on the end of hospitalization, next available visit in the clinic or whichever occurred first. Patients’ outcomes were reported at the time of completion of the survey as full recovery, partial recovery, death currently still in hospital or unknown. In addition information about hospitalization (yes/no) and ICU admittance (yes/no) was requested. We have added this explanation to methods section.
Point 7:
Data for the individual factors must be given and also the analyses should be done in the individual groups. What are risk factors for mortality or treatment at the ICU ? Number of deaths and causes of deaths should be given,The entry criteria for ICU are quite different in hospitals. In some hospitals everyone on high flow oxygen is admitted to ICU in others only if the are artificially ventilated. The criteria should be described.
Response 7: Thank you for this remark. As already expressed our study is a real world study therefore certain limitations cannot be excluded. However even WHO Solidarity study did not avoid the regional differences in SoC.
Point 8:
For analyzing the role of radiological signs to death, it is important to document the causes of death such as ARDS, myocardial infection, pulmonary embolism, septic shock.
They report that patients with hepatitis C and B infection have more atypical radiological signs. As these viruses don´t produce radiological signs except for very rare autoimmune phenomena, these changes should be described: bacterial pneumonia?
Response 8: Thank you for this remark. We do not have information about causes of deaths or autopsy results. The observed higher proportion of radiological changes in HCV and/or HBV infection is probably by chance distribution, which is common in observational studies and in logistic multivariate regression analysis was approached by adjustment by potential confounding factors showing that HCV/HBV co-infection is not an independent risk factor for poor COVID-19 outcome We made sure this is clearly expressed in the paper.
Point 9:
In summary, I think it is quite disappointing that the authors have an interesting large data base of HIV-infected patients with Covid and they are not able or not willing to present their data in a way that they can address the important questions. They present a quite trivial conclusion that a SARS-CoV-2-infected patient with a pneumonia has a worse prognosis in comparison to patients without a pneumonia. Instead, they should evaluate their cohort regarding the risk factors for a more severe course of infection, for a worse prognosis, for treatment on the ICU and for death.
Response 9: Our study team consists of over twenty infectious diseases specialists being recognized experts in the field of HIV and COVID-19. ECEE group has intensively published in many peer reviewed medical journals. We do not share the opinion of the reviewer on what is important and how we should analyze this dataset.
Point 10:
If atypical radiological signs are caused by other infections, the role of coinfections should be addressed.
Response 10: Thank you for this remark. We did not collect information on microbiological diagnoses therefore it was not discussed in the paper.
Point 11:
If specific radiological signs (and not only the extent of lung involvement) are linked to a worse prognosis, then, this should be reported, but the atypical radiological signs must be presented in detail because a wide range of radiological signs are considered atypical.
Radiological signs are a symptom of a more severe Covid-19. If the risk factors for a more severe Covid-19 are evaluated, then it is important to know when the radiological examination occurred in the course of the stay in the hospital and which of consecutive examination was included in the analysis.
Response 11: Thank you for this remark. We aimed to analyze presence of radiological changes and their association with COVID-19 outcomes among HIV-positive individuals. We decided to divide the variable: radiological changes into three categories only: typical, atypical and no radiological changes. However, the idea of performing another analysis regarding more detailed division of radiological changes is very valuable and worth considering for future studies.
Point 12:
If patients with hepatitis B/C have a more severe course of infection, then it should be investigated whether this is caused by hepatitis B/C infection itself or by hepatic insufficiency or cirrhosis.
Response 12: Thank you for this remark. Please see out comments to Point 8.

Round 2
Reviewer 2 Report
The authors resubmit a revised manuscript. Unfortunately, they addressed only some of my concerns but not the main point:
I am quite astonished that a „ study team of recognized experts in the field of HIV and Covid-19“ resubmits a manuscript with such a trivial conclusion that someone with a respiratory infection has a higher risk of progression if there is a radiological proven pneumonia in comparison to patients without radiological symptoms. This conclusion is trivial for Covid-19 , with and without concomitant HIV infection.
It is very disappointing that the group has plenty of interesting data and nevertheless they stress such an trivial aspect. If their real world data base cannot provide detailed information about radiological signs and analyses, then they should restrict their analysis to aspects they can address.
The authors do not understand that patients with radiological signs have a more severe disease than patients without radiologic signs. Instead of presenting the trivial conclusion that patients with a more severe disease have a worse outcome, they should use their database to focus on the risk factors for a severe outcome in HIV-infected patients with covid. They provide information regarding the risk factors already in table 2, but they should describe them in more detail such as how many of their patients needed ICU treatment and how many died of Covid-19 and what were the factors that patients got hospitalized, were on the ICU ward or died.
If their database does not provide essential information about the cause of death (HIV, Covid-19, pneumonia, sepsis, ..), then they should rethink their current database.
I would propose that they change the focus of their manuscript to the „Analysis of risk factors for a severe course of Covid „ and leave the radiology out of the title . They can use radiological signs for the grading of the severity of disease and include them as in table 2, but the focus must be on the analysis of factors being associated with a severe course of disease (death or ICU ward). The outcome partial recovery is not useful if not more information is given (partial recovery after which time, extent of disease).
Furthermore, the answers to several questions are insufficient:
Point 2:
In the introduction they state that „the aim of the study was to analyze radiological changes and their association with covid-19 outcomes a…“, however, they do not provide a detailed analysis of radiological changes. They only indicate that a one group of patients had any type of bilateral consolidations and another group had atypical changes without giving more detailed information about the extent of radiological signs or giving information of the kind of atypical signs. It makes a big difference whether a person has an enlarged lymph node or a pleural effusion. I cannot see an „ analysis of radiological changes“ as the authors claim. This analysis is absolutely insufficient with regard to the radiological signs. If their documented data do not allow are more detailed radiological analyses, then, they should not submit a paper with a focus on radiology.
Point 3:
The response is quite remarkable. When the authors write: „ It is a well established practice to treat tuberculosis based on radiological picture even in lack of microbiological examination“ . I hope for the sake of the patients of the participating hospitals that it is clinical standard to perform a bronchoalveolar lavage to get a microbiological diagnosis of mycobacterial infections or pneumocystis or other bacterial infections. If they are not able to perform state of the art diagnosis of opportunistic infections, then they should not submit manuscripts dealing with such topics. It does not make sense to form a patient group with atypical radiological signs without having a clue whether these changes are induced by Covid-19 or by other infections.
Point 6:
The authors did not address this point sufficiently:
They write: „Patients’ outcomes were reported at the time of completion of the survey as full recovery, partial recovery, death currently still in hospital or unknown.“
In the manuscript they describe a composite outcome with death, ICU admission or partial recovery. In the previous submission they write instead of partial recovery no improvment. Furthermore in the method section line 130 they write again no improvement instead of partial recovery. There is quite a difference between partial recovery and no improvement and the authors should explain why the changed this. I think the methods section is quite confuse. In addition, I do not understand why a partial recovery or even no improvement is combined with ICU admission or death. This does not make sense, especially when the clinical symptoms are not graded and if there are no time lines provided. Partial or full recovery after which time period? Full recovery after 4 weeks or partial recovery after one week? Partial recovery from a sore throat or from a critical care polyneuropathy?
Therefore, numbers must be given: How many patients were admitted to the ICU ward and how many died. They should be compared to patients who were never hospitalized and to patients who were never treated on the ICU ward.
Point 7.
It is quite annoying to get such an answer to a clear questions: If their „real world study“ cannot address basic questions then the authors should focus on questions they can address.
Point 10:
The authors discuss in the introduction (line 86-89) that opportunistic infections may influence the course of Covid-19 and radiological signs, but they do not provide information on concurrent opportunistic infections. They admit in their response that they did not even collect microbiological diagnosis in their database. I am wondering what is the value of a database for HIV-infected subjects with Covid-19 when they don´t even gather information about ímportant concomitant opportunistic infections.
Point 11:
I think the conduct of a more detailed analysis of radiological changes is not only valuable but mandatory for a publication dealing with radiological aspects of Covid-19.